# Fabrication of Micro-Patterned Surface for Pool-boiling Enhancement by Using Powder Injection Molding Process

**DOI:** 10.3390/ma12030507

**Published:** 2019-02-07

**Authors:** Hanlyun Cho, Juan Godinez, Jun Sae Han, Dani Fadda, Seung Mun You, Jungho Lee, Seong Jin Park

**Affiliations:** 1Department of Mechanical Engineering, Pohang University of Science and Technology, 77 Cheongam-ro, Nam-gu, Pohang, Gyeongsangbuk-do 37673, Korea; forever1246@postech.ac.kr; 2Department of Mechanical Engineering, The University of Texas at Dallas, 800 W. Campbell Rd., Richardson, TX 75080, USA; juan.godinez@utdallas.edu (J.G.); fadda@utdallas.edu (D.F.); you@utdallas.edu (S.M.Y.); 3Department of Nano Manufacturing Technology, Korea Institute of Machinery and Materials, 156 Gajeongbuk-ro, Yuseong-gu, Daejeon 34103, Korea; jshan@kimm.re.kr; 4Department of Energy Conversion Systems, Korea Institute of Machinery and Materials, 156 Gajeongbuk-ro, Yuseong-gu, Daejeon 34103, Korea; jungho@kimm.re.kr

**Keywords:** powder injection molding, copper micro-pattern, pool boiling, critical heat flux, heat transfer coefficient

## Abstract

In this study, two kinds of copper micro-patterned surfaces with different heights were fabricated by using a powder injection molding (PIM) process. The micro-pattern’s size was 100 μm, and the gap size was 50 μm. The short micro-pattern’s height was 100 μm, and the height of the tall one was 380 μm. A copper powder and wax-polymer-based binder system was used to fabricate the micro-patterned surfaces. The critical heat flux (CHF) and heat transfer coefficient (HTC) during pool-boiling tests were measured with the micro-patterned surfaces and a reference plain copper surface. The CHF of short and tall micro-patterned surfaces were 1434 and 1444 kW/m^2^, respectively, and the plain copper surface’s CHF was 1191 kW/m^2^. The HTC of the plain copper surface and the PIM surface with short and tall micro-patterned surfaces were similar in value up to a heat flux 1000 kW/m^2^. Beyond that value, the plain surface quickly reached its CHF, while the HTC of the short micro-patterned surface achieved higher values than that of the tall micro-patterned surface. At CHF, the maximum values of HTC for the short micro-pattern, tall micro-pattern, and the plain copper surface were 68, 58, and 57 kW/m^2^ K.

## 1. Introduction

The surface of a reactor in a nuclear power plant requires highly effective cooling. It is cooled by water that can boil under atmospheric pressure at the surface [1], protecting the surface from reaching excessive temperatures, which can compromise the integrity of the reactor vessel. Achieving effective boiling heat transfer is required in order to improve the nuclear power plant’s safety and prevent damage that can occur due to excessive heat. 

Research in pool-boiling heat transfer can be directly used in this cooling application, among numerous other applications. Representative quantities for pool-boiling heat transfer are the critical heat flux (CHF) and the heat transfer coefficient (HTC) [2]. CHF is the upper limit of the nucleate boiling regime where a vapor film blankets the heating surface and causes a significant reduction of the HTC. When the applied heat flux increases towards CHF, vapor generation prevents the surface from being wetted effectively. If the surface is not wetted, boiling at the surface is obstructed and the surface temperature increases dramatically. CHF is defined as heat flux just before the dramatic increase of surface temperature. It is the maximum heat flux where effective boiling can occur at a hot surface. Achieving high CHF values allows effective heat dissipation in high heat flux applications. 

Meanwhile, HTC is the ratio of the heat flux at the surface to the temperature difference between the heating surface and the working fluid. In the nuclear reactor application, achieving a high HTC results in effective removal of heat while keeping the surface at temperatures within the design limits of the vessel. 

Researchers have studied how CHF and HTC can be increased by fabricating porous coatings on a plain surface [1,3,4,5,6,7,8,9,10], nano-structures on a plain surface [11,12], laser-processed structures [4,13], rough surfaces [14,15], enhanced designed surfaces [16], micro-patterned structures [6,15,17,18,19,20], or combined structures with micro-patterned structures and porous coating [6,21,22]. Their results indicate significant improvements in both the CHF and HTC. Common fabrication methods for the micro-pattern or micro-channel are electric discharge machining [6,17] or computer numerical control (CNC) machining [21,22] for metallic surfaces and dry etching [15,18,19,20] for silicon surfaces.

In this paper, the effects of micro-patterns on CHF and HTC with distilled water on a copper surface are investigated. The micro-patterns were fabricated using a powder injection molding (PIM) process, because complex-shaped structures can be easily fabricated with low production costs by using the PIM process [23]. The PIM process facilitates the mass-production of the micro-patterns with low production costs, and it is the most useful advantage compared with other common fabrication methods [24]. The micro-patterned surface was fabricated with material and a structure suitable for pool-boiling heat transfer. Copper powder was used to fabricate the structure due to copper’s high thermal conductivity. The structure was fabricated with small gaps between the patterns, since small gaps lead to high capillary forces [25]. 

Various researchers have also studied powder injection molded (PIMed) micro-patterns [24,26,27,28,29,30]. All of the PIMed micro-patterns were fabricated by using a sacrificial mold with a reversed shape of the pattern. Silicon [26,28] or polymethylmethacrylate (PMMA) [24,29,30] were used for the sacrificial mold. The reversed shape of the micro-pattern was fabricated by using deep reactive ion etching for silicon, or X-ray lithography for PMMA. The micro-patterns with the silicon sacrificial mold usually had a small pattern size and high aspect ratio, but had a large gap compared to the micro-patterns with the PMMA sacrificial mold. However, the researchers usually developed the PIMed micro-patterns not for heat transfer applications. but sensor and actuator applications. The introduction of micro/nano structures on a heating surface have been found to enhance liquid spreading on a heating surface and to delay dry-out. In pool-boiling applications, this results in an increase in the HTC and the CHF [19,31].

Tall and short micro-patterns described in this paper are suitable for pool-boiling and exhibit better CHF and HTC than a plain surface.

## 2. Materials and Methods 

### 2.1. Preparation of Materials

Copper powder with purity of 99.9% (supplied by HKK solutions, Seoul, Korea) was used in this research. The particle size and density of the powder are shown in Table 1. Particle size was measured by using a laser-scattering particle-size analyzer (Horiba Partica LA-950V2, Kyoto, Japan). Density was measured by using helium picnometry (Accupyc 1330, Norcross, GA, USA) at room temperature (24 ± 1 °C). Figure 1 shows the powder morphology observed by scanning electron microscopy (SEM, Akishima, Tokyo, Japan). The particles had a spherical shape.

A wax-polymer-based binder system (supplied by CetaTech, Cheongju, Chungbuk, Korea) was used as a binder. The binder consisted of paraffin wax (PW), polypropylene (PP), polyethylene (PE), and stearic acid (SA). Detailed information about the binder is shown in Table 2. Physical properties of the binder were determined from the reference [32].

Sacrificial molds which had a reversed shape of the micro-pattern were fabricated by the same method as our previous work for fabrication of PIMed lead zirconate titanate (PZT) micro-patterns [31]. The polymethylmethacrylate (PMMA) plate was selectively exposed to an X-ray through a mask which had the same shape as the micro-pattern. The exposed area in the PMMA plate was dissolved in GG developer (a chemical mixture of 60 vol. % 2-(2-butoxyethoxy) ethanol, 20 vol. % tetrahydro-1, 4-oxazine, 5 vol. % 2-aminoethanol, and 15 vol. % deionized water) [31,33]. Two kinds of PMMA plates with different thicknesses were used. Thus, PMMA sacrificial molds with different thicknesses but identical pattern sizes were fabricated.

### 2.2. PIM Process

A common PIM process consists of four steps: mixing, injection molding, debinding, and sintering [23]. In the mixing step, the volume fraction of the copper powder to the total volume of the powder and the binder system is called solid loading. Critical solid loading was evaluated by measuring mixing torque at various solid loading ranges with a HAAKE PolyLab QC. A dramatic increase in mixing torque occurs after critical solid loading [32]. The copper powder and the binder system were mixed with slightly less solid loading than the critical solid loading in order to facilitate the injection molding process [32]. The mixing process was conducted with a twin-extruder mixer at 160 °C. The mixture of the powder and the binder system is called a feedstock. 

After the mixing process was complete, the injection molding process was conducted in order to shape the micro-pattern. Sodick Plustech TR30EH (Yokohama, Kanagawa, Japan) was used for the injection molding process. The PMMA sacrificial mold was placed in a rectangular mold cavity as a mold insert. The feedstock was injected at 160 °C. The temperature of the mold was 45 °C, the plunger’s injection speed was 30 mm/s, and the maximum filling pressure was 80 MPa. 

Next, the debinding process in this research consisted of three steps: demolding, solvent debinding, and thermal debinding. The demolding step was conducted to remove the PMMA sacrificial mold by immersing the injected specimens in acetone at 50 °C, so that the sacrificial mold was dissolved in acetone. Then, the injected specimen had micro-patterns. PW and SA were removed during the solvent debinding step. The specimens were immersed in N-hexane at 50 °C, and both PW and SA were dissolved in N-hexane. Then, PP and PE were removed during thermal debinding by thermal degradation in the tube furnace (Kejia KJ-1600G, Zhengzhou, China). The thermal debinding was conducted with two holding stages in a hydrogen atmosphere. The first holding stage was at 250 °C for three hours in order to remove the remaining PW and SA from the solvent debinding step. The second holding stage was at 450 °C for three hours in order to remove PP and PE. Because the micro-pattern had a high aspect ratio and small pattern size, the structure could be deformed with a high heating rate during the thermal debinding step [31,34]. Therefore, the feedstock was heated slowly at a rate of 0.5 °C/min. Finally, the specimens were sintered at 700 °C in a hydrogen atmosphere. The sintering process was conducted in a tube furnace (Kejia KJ-1600G) with a heating rate of 0.5 °C/min.

### 2.3. Pool-Boiling Experiment

A heater assembly with a surface area of 6.3 mm × 19 mm was fabricated for pool-boiling experiments using distilled water. The assembly was used in a pool-boiling chamber at atmospheric pressure. Schematics of the heater assembly and the chamber are shown in Figure 2. The heater assembly consisted of a micro-patterned surface, a machined copper block with a surface area of 6.3 mm × 19 mm and 3 mm depth (copper purity of 99.99%, supplied by McMaster-Carr, Elmhurst, IL, USA) with two holes along the 19 mm × 3 mm side that housed thermocouples (30 AWG T-Type with a diameter of 0.25 mm), and three 6.3 mm × 6.3 mm heaters (20-ohm resistance heaters with a maximum power of 350 Watts, supplied by Component General, Inc., Odessa, FL, USA). A heater assembly without the micro-patterned surface was also prepared as a control test. The machined copper block with its thermocouples was attached below the micro-patterned surface by using a Rosin core 97/3 lead-free solder. The heaters were also attached below the copper block using the same solder. When the control heater assembly (without micro-patterned copper surface) was prepared, only the heater and machined copper block were attached to each other using the solder. In this case, the heating surface of the plain copper block was prepared by sanding and its surface roughness was 0.35 µm Ra. All other surfaces except the top surface were insulated with epoxy (3M DP420). A polycarbonate block was attached at the bottom of the heater, using the same epoxy, in order to fix the heater assembly in the test chamber.

The thermal conductivities of epoxy (0.2 W/m·K) and polycarbonate (0.2 W/m·K) were sufficiently lower than that of copper. Therefore, heat transfer in the heater assembly is represented by one dimensional conduction [1]. The surface temperature of the micro-pattern structure was calculated from the measured temperature by using Equation (1). The terms representing the solder and PIM are excluded from Equation (1) for the tests with a plain copper surface:(1)$$T_{s}=T_{m}-\left(\frac{\frac{1}{2}t_{block}}{k_{block}}+\frac{t_{solder}}{k_{solder}}+\frac{t_{PIM}}{k_{PIM}}\right)q^{\prime \prime },$$
where *T_s_* is the surface temperature, *T_m_* is the average of the measured temperatures from the thermocouples in the middle of the copper block, *t_block_* (3.0 mm) is the thickness of the machined copper block, *k_block_* (401 W/m-K) is the thermal conductivity of the machined copper block, and *t_solder_* (70 µm for the assembly with the short micro-pattern and 90 µm for the assembly with the tall micro-pattern) is the thickness of the solder, and is obtained by measuring the overall thickness of the assembly minus the thicknesses of the individual components. *k_solder_* (77 W/m-K) is the thermal conductivity of the solder which was provided by the manufacturer, *t_PIM_* (1.53 mm for the PIMed block with the short micro-pattern and 1.48 mm for that with the tall PIMed micro-pattern) is the total thickness of the PIMed copper substrate and its micro-pattern, and *k_PIM_* (78 W/m-K) is the thermal conductivity of the PIMed copper substrate and was obtained from a one-dimensional conduction heat transfer experiment. Constant properties are assumed for the physical properties. *q″* is the provided input heat flux, which is the supplied power to the heater, divided by the area of 6.3 mm × 19 mm. 

When the test was conducted without the micro-patterned surface, the surface temperature of the plain copper surface could be calculated by using Equation (2).
(2)$$T_{s}=T_{m}-\frac{\frac{1}{2}t_{block}}{k_{block}}q^{\prime \prime }.$$

The nucleate boiling heat-transfer coefficient (HTC) can be obtained from Equations (3) and (4).
(3)$$HTC=\frac{q^{\prime \prime }}{\Delta T_{sat}},$$
(4)$$\Delta T_{sat}=T_{s}-T_{sat},$$
where *ΔT_sat_* is the wall superheat, and *T_sat_* is the saturation temperature of the water, respectively.

Each heater assembly was immersed in distilled water in the test chamber. The water was degassed for 45 min by using an immersion heater, and the temperature of the water was maintained at saturation temperature by using band heaters. Power was applied to the test heater to generate the heat flux as the pool-boiling experiment was commenced. 

During the pool-boiling experiment, a stepwise increase in the power to the heater assembly was imposed by the data acquisition system until the heater reached CHF. The data acquisition program holds a constant power supply to the heater assembly at each step to establish a steady state. The program collects data and calculates the wall superheat *ΔT_sat_* every 250 ms and compares a 20 s running average to the prior 20 s average of *ΔT_sat_*. Steady state is determined when the difference between the averages is less than 0.1 K. CHF is achieved when the instantaneous *ΔT_sat_* suddenly increases to exceed the prior 20 s running average by 10 K. At that step, the CHF value is recorded as the highest heat flux which yields a steady temperature, plus half the added heat flux which caused the sudden increase in *ΔT_sat_*. Then, CHF of the specimen is defined as the maximum heat-flux value recorded during the experiment.

A LabVIEW program was used to collect the data and to control the pool-boiling experiment. The LabVIEW program was also used to control the direct current (DC) power supply (Agilent N5771a, Santa Rosa, CA, USA) and the data acquisition system (Agilent 34980A Multifunction Switch/Measure Module equipped with an Agilent 34921T 40-Channel Armature Multiplexer, Santa Rosa, CA, USA). The program calculates the heat flux by measuring the voltage and current at the resistance heater.

### 2.4. Experimental Uncertainties

The experimental uncertainties were calculated using the single-sample experimental method developed by Kline and McClintock [35]. The calculated wall superheat uncertainty was within ±0.6 K for the range of parameters used in this work. The calculated heat flux uncertainty was ±31 kW/m^2^ at a heat flux of 1500 kW/m^2^. For the range of heat flux used in this work, the heat flux, and thermal conductivity of the PIMed surface, *k_PIM_*, uncertainty was consistently within ±2.5%. 

## 3. Results and Discussion

### 3.1. Fabrication of the Micro-Patterned Surface

The mixing torque of feedstock at a solid-loading range from 51% to 61% by volume is shown in Figure 3. Beyond a solid-loading of 58% by volume, the torque was found to increase rapidly, so the critical solid loading was defined as 58% by volume.

The feedstock was fabricated with solid loading of 56% by volume. The fabricated feedstock and PMMA sacrificial mold are shown in Figure 4. The fabricated PMMA mold had a uniform pattern size and gap size.

The PMMA sacrificial mold was placed in a rectangular mold cavity, shown in Figure 5. Feedstock was injected into the mold. The injected specimen attached to the PMMA mold is shown in Figure 6. The transparent part in Figure 6 is the PMMA mold. Feedstock was completely packed into the empty space of the PMMA mold. The PMMA mold and the binder system were removed during the debinding step. Finally, sintered micro-patterned surfaces were fabricated after the sintering process. SEM images of the micro-patterns are shown in Figure 7. Straight and uniform micro-patterns were fabricated. All dimensions were measured from the SEM images. 

The pattern size is 100 μm and the gap size between the patterns is 50 μm. The short pattern’s height is 100 μm, and the tall pattern’s height is 380 μm. An additional surface was also fabricated using this same PIM method with no micro-patterns. 

### 3.2. Contact Angle of the Micro-Patterned Surface

In order to characterize the wetting behavior of the micro-patterned surface, the apparent contact angle was measured by using a goniometer (Krüss DSA30, 0.3° measurement accuracy, Hamburg, Germany). The surface was cleaned using 5% acetic acid in an ultrasonic bath prior to the contact angle measurements. Then, the surface was rinsed by distilled water and dried by an air jet. A single water droplet of 11.5 mm^3^ was dripped onto the surface. The contact angle measurement was repeated three times, and the average of the three results is reported.

Contact angles of the short-patterned, tall-patterned, and plain copper surface fabricated by PIM with no micro-patterns are 11.7°, nearly 0°, 64.9°, and 35.1°, respectively (shown in Figure 8). 

The contact angle on the PIMed copper surface with no micro-patterns is significantly less than that of a plain copper surface. The contact angles on the micro-patterned surfaces are observed to decrease further. Of particular note is the tall-patterned surface, which appears fully wetted within 0.161 s after the water droplet contacts the surface. This apparent decrease in the contact angle is partially due to the reduced contact angle of the PIMed material and also partially due to the patterned surface behaving as a pin wall which allows water to penetrate between the pins during the measurements.

### 3.3. Results of the Pool-Boiling Experiments

The top surface of the test heater assembly during the test at 50 kW/m^2^, 500 kW/m^2^, and 1000 kW/m^2^ is shown in Figure 9 for the plain copper surface, the short micro-patterned surface, and the tall micro-patterned surface.

Results of the pool-boiling experiments are shown in Figure 10 in terms of the pool-boiling curve (a) and the HTC curve (b). The CHF of the plain copper surface is measured as 1191 kW/m^2^. The CHF of the short and tall micro-patterned surfaces were measured as 1434 and 1444 kW/m^2^, respectively. The CHF value for either micro-patterned surface exhibits a 20% improvement over that of a plain copper surface. This increase is due to the reduced contact angle [11,36] which allows the surface to be easily and continuously wetted and delays dry-out of the surface.

HTC for the PIMed surfaces and the plain copper surface are similar in value when the applied heat flux is below 1000 kW/m^2^. However, HTC of the short micro-patterned surface is found to exceed that of the tall micro-patterned surface when the applied heat flux is increased beyond 1000 kW/m^2^. The maximum values of the HTC for the short micro-pattern, tall micro-pattern, and the plain copper surface, based on the same plain copper area, are 68, 58, and 57 kW/m^2^·K. The CHF values for the PIMed surfaces exhibit an increase of 19% and 2% over the value for a plain copper surface for the PIMed surface with the short and tall micro-patterns, respectively.

The decrease in the HTC on the tall micro-patterned surface relative to the short micro-patterned surface is due to more vapor (dry spots) contained within the thicker PIMed surface. It seems that generated vapor may be trapped among the tall micro-patterns, and the trapped vapor reduces the boiling heat-transfer on the surface. 

The true surface areas of the boiling surfaces (base plus all the surfaces of pins) were calculated for the short and tall micro-patterns over the entire surface. The extended areas for the short and tall micropatterns were 79% and 301% larger than the area of the flat plain surface area. This percentage increase in area is much larger than the improvement in CHF (20% improvement in CHF with micropatterns) and HTC (19% for the short and 2% for the tall micro-patterns at CHF). 

## 4. Conclusions

Two kinds of copper micro-patterned surfaces with different heights were successfully fabricated by using the powder injection molding process. Boiling heat-transfer tests with distilled water at an atmospheric pressure were conducted using these surfaces. The micro-patterns enhance boiling heat-transfer. Specifically, the micro-patterned surfaces exhibit an enhancement in CHF by 20% over a plain copper surface. The HTC values using the tall micro-patterned surface and the short micro-patterned surface are comparable to one another and to the plain copper surface at heat flux values below 1000 kW/m^2^. The HTC with a short micro-patterned surface was shown to exhibit an advantage over the tall micro-patterned surface at heat-flux values over 1000 kW/m^2^.

## Figures and Tables

**Figure 1 materials-12-00507-f001:**
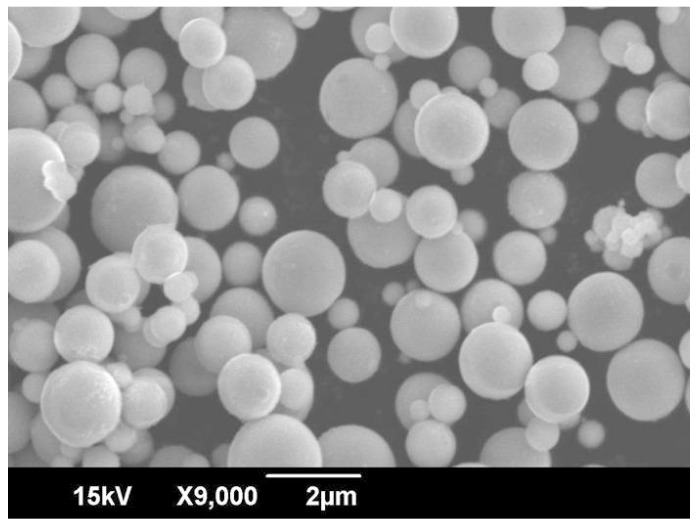
Scanning electron microscopy (SEM) image of the copper powder.

**Figure 2 materials-12-00507-f002:**
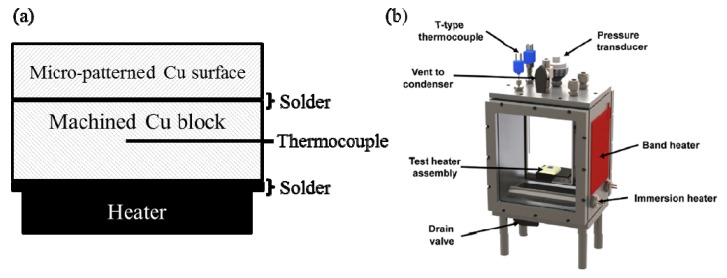
(**a**) Schematics of test heater assembly (schematic not to scale); (**b**) test chamber [1].

**Figure 3 materials-12-00507-f003:**
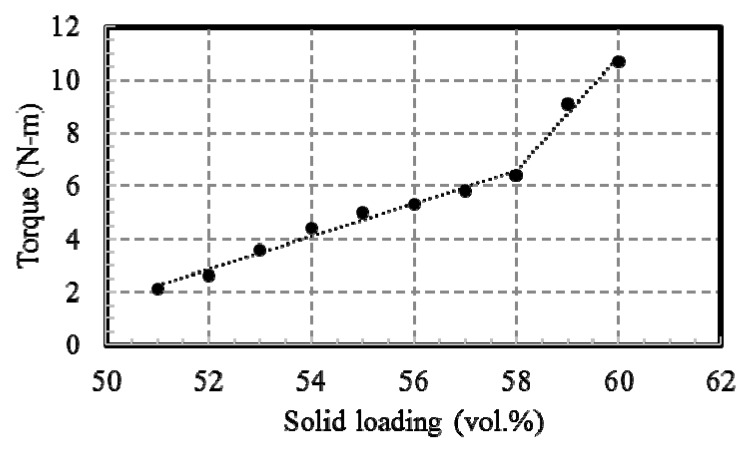
Results of the mixing torque measurement.

**Figure 4 materials-12-00507-f004:**
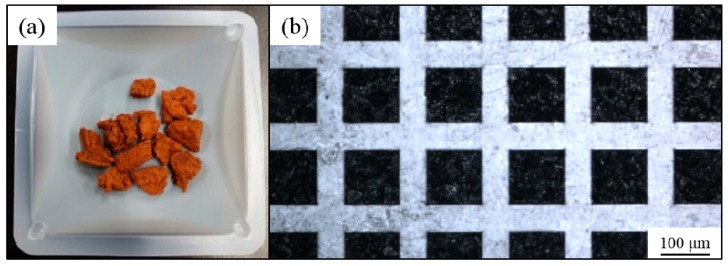
(**a**) Fabricated feedstock; (**b**) polymethylmethacrylate (PMMA) sacrificial mold.

**Figure 5 materials-12-00507-f005:**
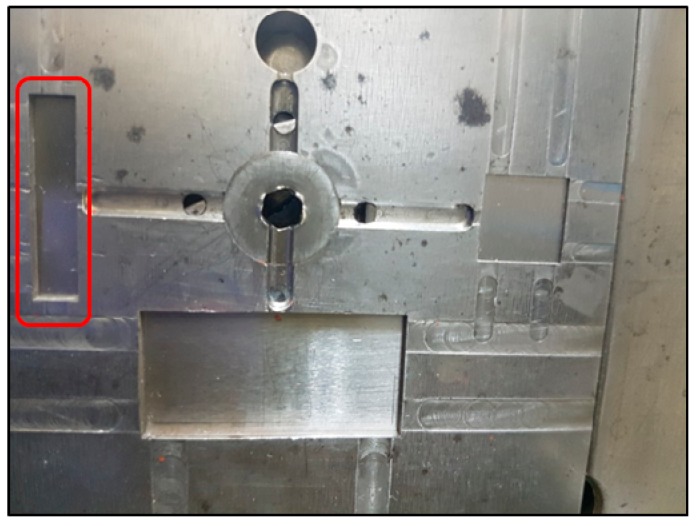
Rectangular mold cavity (inside the red rectangle).

**Figure 6 materials-12-00507-f006:**
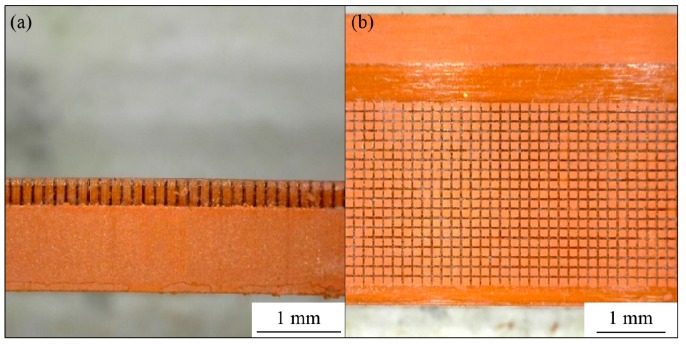
Injected specimen attached with PMMA sacrificial mold: (**a**) side view; (**b**) top view.

**Figure 7 materials-12-00507-f007:**
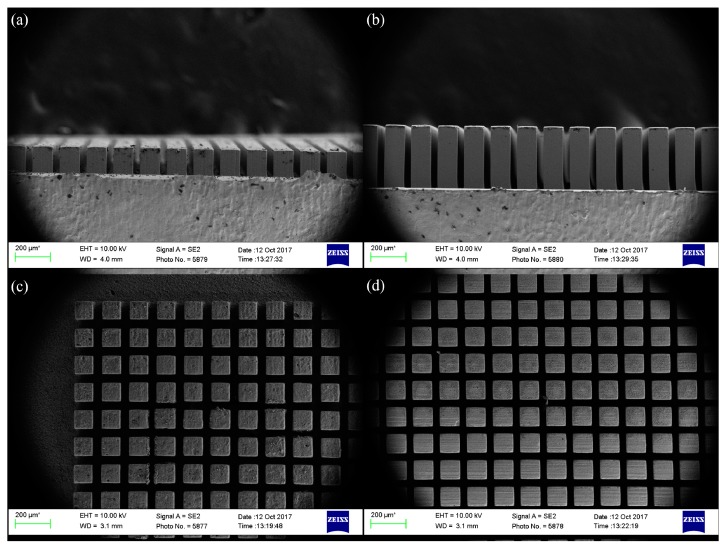
SEM images of sintered micro-patterns: (**a**) side view of the short micro-pattern; (**b**) side view of the tall micro-pattern; (**c**) top view of the short micro-pattern; and (**d**) top view of the tall micro-pattern.

**Figure 8 materials-12-00507-f008:**
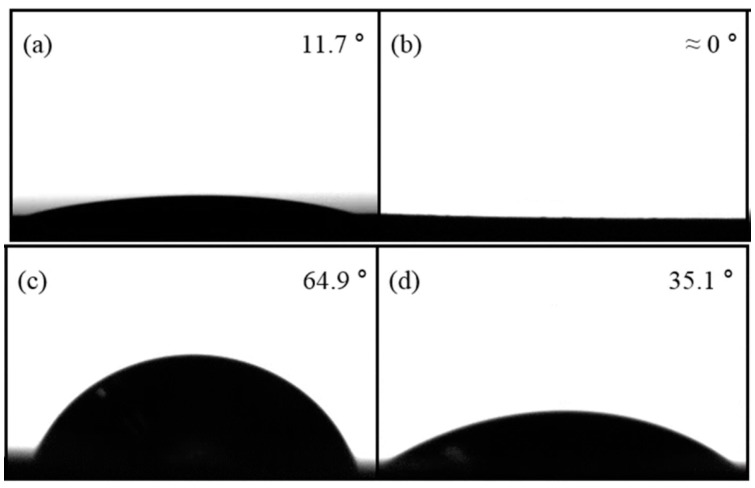
Contact angles of each surface: (**a**) the short micro-patterned surface; (**b**) the tall micro-patterned surface; (**c**) the plain copper surface; and (**d**) the flat copper surface fabricated by powder injection molding (PIM) with no micro-patterns.

**Figure 9 materials-12-00507-f009:**
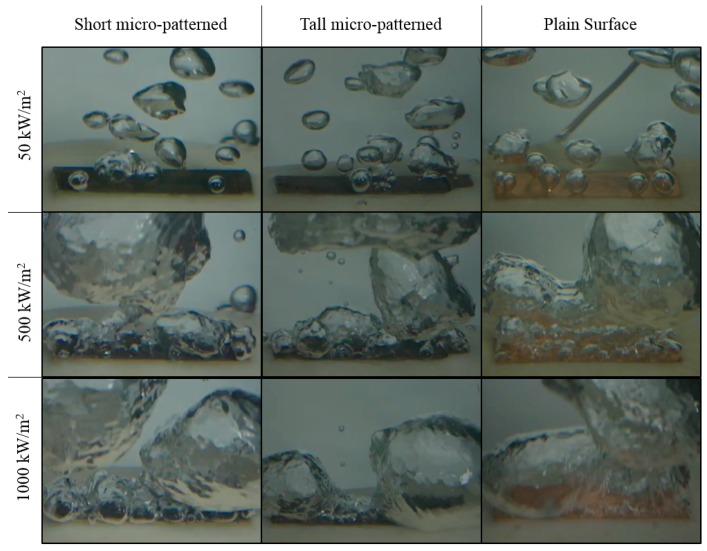
Captured images during the pool-boiling tests.

**Figure 10 materials-12-00507-f010:**
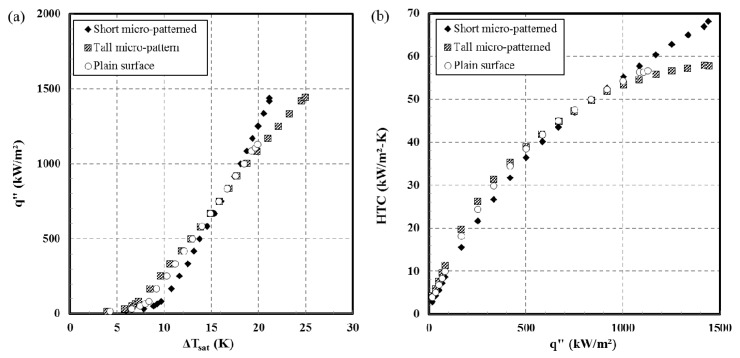
Boiling curves (**a**) and nucleate boiling heat-transfer coefficient (**b**) of the micro-patterned surfaces and plain surface.

**Table 1 materials-12-00507-t001:** Particle size and density of the copper powder.

*D*_10_ (μm)	*D*_50_ (μm)	*D*_90_ (μm)	Density (g/cm^3^)
2.01	4.73	15.23	8.638

**Table 2 materials-12-00507-t002:** Wax-polymer-based binder system.

Component	Contents (wt. %)	Melting Point (°C)	Decomposition Range (°C)	Density (g/cm^3^)
Paraffin wax (PW)	57.5	51	242–280	0.92
Polypropylene (PP)	25.0	78	464–481	0.92
Polyethylene (PE)	15.0	120	464–471	0.93
Stearic acid (SA)	2.5	53	246–275	0.95

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
