# Peer review of "Fabrication of Micro-Patterned Surface for Pool-boiling Enhancement by Using Powder Injection Molding Process"

_materials, 2019, doi:10.3390/ma12030507_

Reviewer 1 Report

I suppose that your paper is worthwhile for research of “Fabrication of Micro-Patterned Surface for Pool Boiling Enhancement by Using Powder Injection Molding Process”. However, I have some questions about your paper.

 #1 on Page 4, around line 137-146.

You are expressed the equation of calculation for surface temperature, Ts. I would like to know the dimensions of thickness of the block, solder and PIM, and the values of thermal conductivity of these. Can you add these values in this paragraph or figure. 2 (a)?

In addition, I did not find the calculation method of heat flux q” around this paragraph. Did you use the input (supplied) power of the heater for calculation of the heat flux?

 #2 on Page 4, Eq. (1)-(4)

When you discuss these calculated values using measured values such as temperatures, thicknesses and heat fluxes, it is necessary to do the error evaluation of each equations. How much accuracy does HTC and wall superheat on Eq. (3) and (4)?

 #3 on Page 4, line 156-158

How to decide the CHF condition? Please describe the method of the confirmation of CHF.

 #4 on Page 8, Figure 9.

The name of test piece in captions on Fig. 9 are different from previous one. The name in previous figures are “short” and “tall” micro-patterned. Can you make the test piece name agree all over the figures?

In addition, the size of heat transfer surface in these figures are different. Then, I could not compare the bubble sizes departed from heat transfer surface to use these figure. Can you make uniform the surface size or add the scale bar?

 #5 on Page 8, Line 207-212.

I understood the advantage using the short micro-pattern fabrication for pool nucleate boiling at different three surfaces. Incidentally, how long will heat transfer improve when you compare the present fabrication with similar fabrication of other researchers? You may emphasize an advantage of this fabrication method if you can compare it with the data of other researcher’s one or fabrication methods.

 #6 on Page 9, line 215-218 and Page 8, Figure 10.

You discussed the HTC using Fig. 9. However, I think that the discussion for HTC without figure for HTC vs. heat flux around Fig. 9 is difficult. I suppose that you add the figure of HTC vs. heat flux to be easy to discuss the HTC.

 #7 Page 9, line 218-219.

You described about the reason of decreasing the HTC on the tall micro-patterned. I agreed the reason with you. However, at the Figure 9, I could not find the different bubble nucleation condition of short and tall micro patterned. Could you mind you explain the bubble trapping or different boiling phenomena using Fig. 9 or other pictures?

Author Response

Thank you for your kind comments.

Please find the attached our response to your review.

Sincerely,

Seong Jin Park.

Reviewer 2 Report

Please find attached file.

Author Response

Thank you for your kind comments.

Please find the attached our response to your review.

Sincerely,

Seong Jin Park.

Round  2

Reviewer 1 Report

(I do not have more comment for revised one.)

Reviewer 2 Report

After the revision, I support this paper to be published.